# GraspSplats:
# Efficient Manipulation with 3D Feature Splatting

**Mazeyu Ji***, **Ri-Zhao Qiu***, **Xueyan Zou, Xiaolong Wang**
UC San Diego
*Equal contribution
https://graspsplats.github.io

**Abstract:** The ability for robots to perform efficient and **zero-shot** grasping of object parts is crucial for practical applications and is becoming prevalent with recent advances in Vision-Language Models (VLMs). To bridge the 2D-to-3D gap for representations to support such a capability, existing methods rely on neural fields (NeRFs) via differentiable rendering or point-based projection methods. However, we demonstrate that NeRFs are inappropriate for scene changes due to their implicitness and point-based methods are inaccurate for part localization without rendering-based optimization. To amend these issues, we propose GraspSplats. Using depth supervision and a novel reference feature computation method, GraspSplats generates high-quality scene representations in under 60 seconds. We further validate the advantages of Gaussian-based representation by showing that the explicit and optimized geometry in GraspSplats is sufficient to natively support (1) real-time grasp sampling and (2) **dynamic and articulated object manipulation** with point trackers. With extensive experiments on a Franka robot, we demonstrate that GraspSplats significantly outperforms existing methods under diverse task settings. In particular, GraspSplats outperforms NeRF-based methods like F3RM and LERF-TOGO, and 2D detection methods.

**Keywords:** Zero-shot manipulation, Gaussian Splatting, Keypoint Tracking

## 1 Introduction

Efficient zero-shot manipulation with part-level understanding is crucial for downstream robotics applications. Consider a kitchen robot deployed to a new home: given a recipe with language instructions, the robot pulls a drawer by its handle, grasps a tool by its grips, and then pushes the drawer back. To perform these tasks, the robot must **dynamically** understand **part-level** grasp affordances to interact effectively with objects. Recent work, such as [1, 2, 3], explores this understanding by embedding reference features from large-scale pre-trained vision models (e.g., CLIP [4]) into Neural Radiance Fields (NeRFs). However, those methods [2, 1] offer only a static understanding of the scene at the object level and require minutes to train the scene, necessitating costly retraining after any scene changes. This limitation significantly hinders practical applications involving object displacements, or requiring part-level understanding. On the other hand, point-based methods such as [5], which perform back-projection of 2D features, are efficient in feature construction but struggle with visual occlusion and often fail to infer fine-grained spatial relationships without further optimization.

In addition to dynamic and part-level scene understanding, achieving fine manipulation requires the robot to have a strong understanding of both the geometry and semantics of the scene. For this capability to emerge from coarse 2D visual features, further optimization is necessary to bridge the 2D-to-3D gap. NeRF-based methods [6, 2, 1] facilitate this understanding through differentiable rendering. However, NeRFs [7, 6, 2, 1] are fundamentally implicit representations, making them difficult to edit to accommodate scene changes, thus leading to a **static** assumption. To address

8th Conference on Robot Learning (CoRL 2024), Munich, Germany.

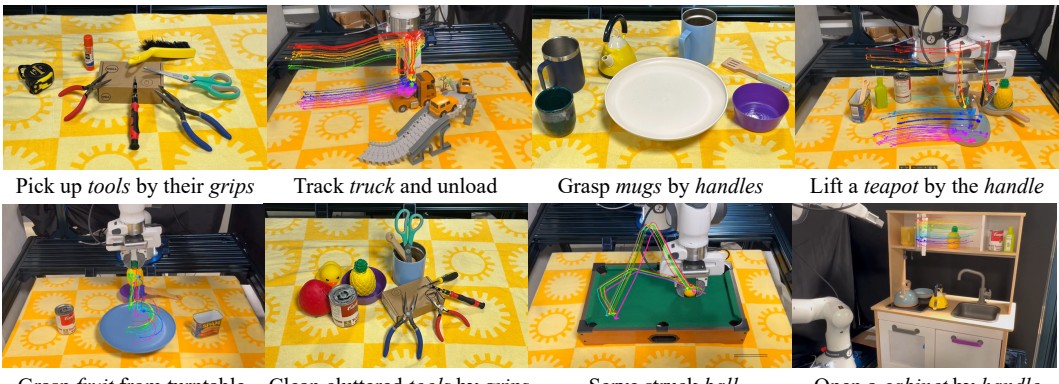

| Pick up *tools* by their *grips* | Track *truck* and unload | Grasp *mugs* by *handles* | Lift a *teapot* by the *handle* |

| Grasp *fruit* from turntable | Clean cluttered *tools* by *grips* | Serve struck *ball* | Open a *cabinet* by *handle* |

Figure 1: **GraspSplats** supports diverse robotics tasks using feature-enhanced 3D Gaussians. Compared to existing NeRF-based methods [1, 2], GraspSplats transforms the feature representation to reflect object motions in real-time with point tracking from one or more cameras, which makes it possible to perform **zero-shot dynamic and articulated object manipulation by parts**.

dynamic problems, some works [8, 9, 10, 11] commonly predict grasp poses using 3D dense correspondences, where reliable grasps are identified based on keypoints in a reference state and then applied to various viewpoints or object placements. However, these methods face challenges in tracking object states over time and handling identical objects.

To this end, we propose GraspSplats. Given posed RGBD frames from a calibrated camera, GraspSplats constructs a high-fidelity representation as a collection of explicit Gaussian ellipsoids via 3D Gaussian Splatting (3DGS) [12, 13]. GraspSplats reconstructs a scene in under 30 seconds and supports efficient part-level grasping for static and rigid transformation, which enables manipulation such as tracking part objects that is not possible with existing methods. GraspSplats initializes Gaussians from the coarse geometry of depth frames; while computing reference features for each input view in real time using MobileSAM [14] and MaskCLIP [15]. These Gaussians are further optimized for geometry, texture, and semantics via differentiable rasterization. The user can supply an object name query (*e.g., 'mug'*) and part query (*e.g., 'handle'*) for GraspSplats to efficiently predict part-level affordance and generate grasp proposals. GraspSplats directly generates grasping proposals using explicit Gaussian primitives in milliseconds, for which we extend an existing antipodal grasp generator [16, 17]. In addition, we further exploit the explicit representation to maintain high-quality representations under object displacement. Using a point tracker [18], GraspSplats coarsely edits the scene to capture rigid transformations and further optimizes it with *partial* scene reconstruction.

We implemented GraspSplats on a desktop-grade computer with a real Franka Research (FR3) robot to evaluate its efficacy in tabletop manipulation. Every component in GraspSplats is efficient and empirically runs a magnitude ($10\times$) faster than existing work [2, 1] — computing 2D reference features, optimizing the 3D representation, and generating 2-finger grasp proposals. This makes it possible to simultaneously generate GraspSplats representation in parallel to arm scans. In experiments, GraspSplats outperforms NeRF-based methods like F3RM and LERF-TOGO, and other point-based methods.

Our contribution is threefold:

- **A framework that advocates 3DGS for grasping representation**. GraspSplats **efficiently** reconstructs scenes with geometry, texture, and semantics supervision, which outperforms baselines on zero-shot part-based grasping in terms of both accuracy and efficiency.

- **Techniques towards an editable high-fidelity representation**, which goes beyond zero-shot manipulation in static scenes into dynamic and articulated object manipulation.

- **Extensive real-robot experiments** that validate GraspSplats as an effective tool for zero-shot grasping in *both static and dynamic scenes*, which demonstrates the superiority of our method over NeRF-based or point-based methods.

## 2 Related Work

**Language-guided Manipulation.** To support zero-shot manipulation, robots must leverage priors learned from internet-scale data. There have been some recent works [19, 20, 21, 22, 23, 24, 5] that use 2D foundation vision models (CLIP [4], SAM [25], or GroundingDINO [26]) to build open-vocabulary 3D representations. However, these methods mostly rely on simple 2D back-projection. Without further rendering-based optimization, they generally fail to provide precise part-level information. Recently, building on LERF [6], researchers [1, 2, 27, 28] have found that combining feature distillation with neural rendering yields promising representations for robotics manipulation, as it offers both high-quality semantics and geometry. Notably, LERF-TOGO [2] proposed conditional CLIP queries and DINO regularization for zero-shot manipulation by parts. F3RM [1] learned grasping from few-shot demonstrations. Evo-NeRF [29] focuses on NeRF specialized for stacked transparent objects. However, these methods are based on NeRFs [7], which is fundamentally implicit. Though certain NeRF representations can be adapted to model dynamic movement, such as grid-based methods [1], dynamic scenes are more natural to be modeled with explicit methods.

**Grasp Pose Detection.** Grasp pose detection has been a long-standing topic [30, 31, 32, 33, 16, 17, 34, 35, 36] in robotics manipulation. Existing methods can be roughly divided into two categories: end-to-end and sampling-based approaches. End-to-end methods [32, 33, 35, 36] offer streamlined pipelines for grasp poses and incorporating learned semantic priors (*e.g.,* mugs grasped by the handle). However, these methods often require the testing data modalities (*e.g.,* viewpoint, object category, and transformations) to match training distribution exactly. For instance, LERF-TOGO [2] resolves viewpoint variation of GraspNet [32] by generating hundreds of point clouds for input using different transformations, requiring significant computational time. Sampling-based methods [16, 17], on the other hand, do not learn semantic priors but offer reliable and rapid results when explicit representations are available. In this work, we find that the *explicit Gaussian primitives* are natural to be connected to sampling-based methods [16, 17], and features embedded in GraspSplats complements the semantic priors via language guidance. This intuitive combination allows efficient and accurate sampling of grasping poses in dynamic and cluttered environments.

**Concurrent Work.** Some concurrent methods [37, 38, 39, 40, 41, 42, 43] interface Gaussian Splatting [12] with 2D features [37, 38, 39] and robotics applications [40, 41, 42, 43]. GraspSplats is based on Feature Splatting [38] for its engineeringly optimized implementation. Among methods for robotics applications, GaussianGrasper [40] mostly supports object-level queries for static scenes and displaces objects only when the robot arm moves them. On the other hand, SplatSim [43] and ManiGaussian [42] require expert demonstrations for policy learning based on GS, where we investigate zero-shot grasping. Most related to our work, SplatMover [41] also studies zero-shot dynamic grasping. Compared to SplatMover, GraspSplats (1) efficiently constructs feature-enhanced GS under 1 minute; (2) shows the efficacy of grasp planning directly on explicit Gaussians than GraspNet [32]; and (3) adapts point-tracking method to displaces object representations in real-time.

## 3 Efficient Manipulation with 3D Feature Splatting

**Problem Formulation.** We assume a robot with a parallel gripper, a calibrated in-wrist RGBD camera, and a calibrated third-person view camera. Given a scene containing a set of objects, the objective is for the robot to grasp and lift an object via language queries (*e.g., 'kitchen knife'*). Optionally, a part query may be further supplied to specify the part to grasp (*e.g., 'handle'*) for task-oriented manipulation [2]. It is worth noting that, unlike previous works [1, 2], we *do not* assume that the scene is static. Instead, we aim to design a more generalized algorithm where part-level grasping affordance and sampling can be done continuously even with object movement.

**Background.** We render depth $\hat{\mathbf{D}}$, color $\hat{\mathbf{C}}$, and features $\hat{\mathbf{F}}$ following existing work [38, 39, 37]:

$$\{\hat{\mathbf{D}}, \hat{\mathbf{F}}, \hat{\mathbf{C}}\} = \sum_{i \in N}\{\mathbf{d}_i, \mathbf{f}_i, \mathbf{c}_i\} \cdot \alpha_i \prod_{j=1}^{i-1}(1 - \alpha_j), \quad (1)$$

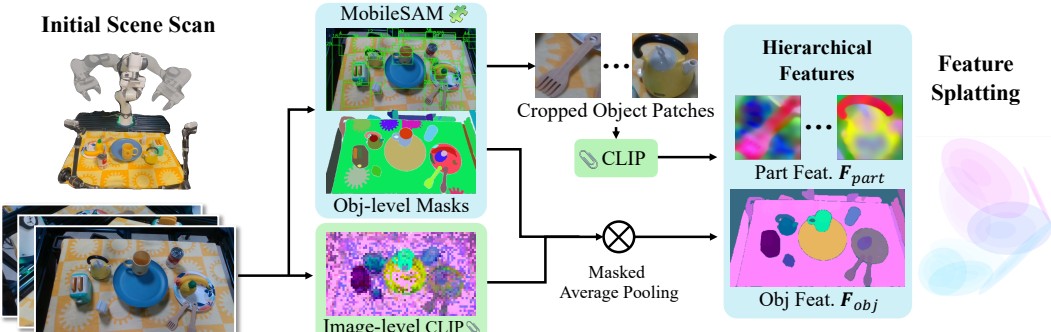

Figure 2: GraspSplats employs two techniques to efficiently construct feature-enhanced 3D Gaussians: hierarchical feature extraction and dense initialization from geometry regularization, which reduces the overall runtime to 1/10 of existing GS methods [38]. (High-dimensional features are visualized using PCA and the visualized Gaussian ellipsoids are trained without densification).

where $\mathbf{d}_i$, $\mathbf{f}_i$, and $\mathbf{c}_i$ is the distance to camera, latent feature, and color. $\alpha_i$ is per-gaussian opacity, and the indices $i \in N$ are ordered by $\mathbf{d}_i$. We provide complete details in appendix Sec. A.

**Overview.** To support open-ended grasping, GraspSplats proposes three key components. The overviews are given in Fig. 2 and Fig. 3. First, a way to construct a scene representation efficiently with novel reference features and geometric regularization. Second, a way to generate grasp proposals directly on 3D Gaussians, using 3D conditional language queries and an extended antipodal grasp proposer [16, 17]. Finally, a way to edit Gaussians under object displacement which enables dynamic and articulated object manipulation.

### 3.1 Constructing Feature-enhanced 3D Gaussians

We use differentiable rasterization [12, 38] to lift 2D features to 3D representation. Though existing works in feature-enhanced GS offers part-level understanding [38, 39], one commonly overlooked weakness is the expensive overhead *before the scene optimization begins*. This overhead can be further dissected to (1) costly reference feature computation [2] or (2) densification of sparse Gaussians [38] originated from SfM preprocessing [12, 44], which we address in this work.

**Efficient Hierarchical Reference Feature Computation.** Existing methods [2, 39, 40] spend most compute efforts to regularize coarse CLIP features — either through thousands of multi-scale queries [2] or mask-based regularization [38, 39, 40] through costly grid sampling [25].

We propose a way to efficiently regularize CLIP using MobileSAMV2 [14]. We generate hierarchical features, object-level and part-level, specialized for grasping. Given an input image, MobileSAMV2 [14] predicts class-agnostic bounding boxes $\mathbf{D}_{obj} := \{(x_i, y_i, w_i, h_i)\}_{i=1}^{N}$ and a set of object masks $\{\mathbf{M}\}$. For object-level feature, we first use MaskCLIP [15] to compute coarse CLIP features of the entire image $\mathbf{F}_C \in \mathbb{R}^{H' \times W' \times C}$. We then follow Qiu et al. [38] and use Masked Average Pooling to regularize object-level CLIP features with $\{\mathbf{M}\}$, which we detail in Sec. F.

For part-level features, we extract image patches from $\mathbf{D}_{obj}$ for batched inference on MaskCLIP [15]. Since $\mathbf{D}_{obj}$ incorporates object priors learned from the SA-1B dataset [25], $N$ is significantly smaller than the number of patches needed from uniform queries [6] for efficient inference. We then interpolate the features to remap them into the original image shape and average over multiple instances to form $\mathbf{F}_{part}$ for part-level supervision.

During differentiable rasterization, we introduce a shallow MLP with two output branches that takes in the rendered features $\hat{\mathbf{F}}$ from Eq. 1 as intermediate features. The first branch renders the object-level feature $\hat{\mathbf{F}}_D$ and the second branch renders the part-level feature $\hat{\mathbf{F}}_{obj}, \hat{\mathbf{F}}_{part} = \text{MLP}(\hat{\mathbf{F}})$,

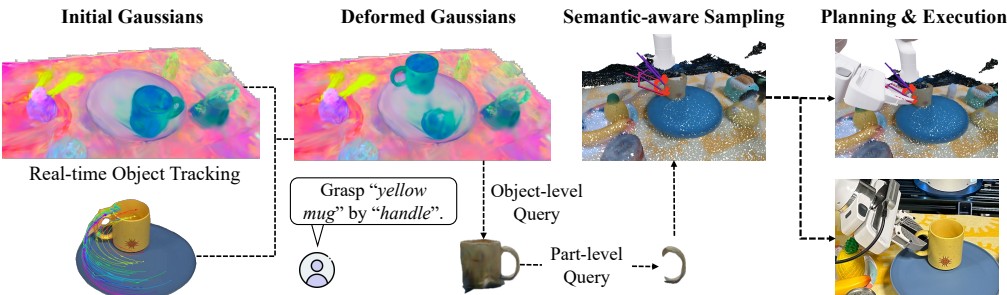

**Initial Gaussians**     **Deformed Gaussians**     **Semantic-aware Sampling**     **Planning & Execution**

Real-time Object Tracking

Grasp "*yellow mug*" by "*handle*".

Object-level Query

Part-level Query

Figure 3: Given an initial state of Gaussians and RGB-D observations from one or more cameras, GraspSplats tracks the 3D motion of objects specified via language, which is used to *deform* the Gaussian representations in real-time. Given object-part text pairs, GraspSplats proposes grasping poses using both semantics and geometry of Gaussian primitives in milliseconds.

where $\hat{\mathbf{F}}_{obj}$ and $\hat{\mathbf{F}}_{part}$ are supervised using $\mathbf{F}_{obj}$ and $\mathbf{F}_{part}$ with cosine loss. We scale the part-level term in the joint loss $\mathcal{L}_{\text{obj}} + \lambda \cdot \mathcal{L}_{\text{part}}$ with $\lambda = 2.0$ to emphasize part-level segmentation.

**Geometry Regularization via Depth.** Existing feature-enhanced GS methods [38, 39, 37] have no supervision for geometry. In GraspSplats, we project points from depth images as centers of the initial Gaussians. In addition, we use depth as supervision during training. Empirically, this additional geometric regularization significantly reduces the training time and better surface geometry.

## 3.2 Static Scene: Part-level Object Localization and Grasp Sampling

To support efficient zero-shot part-level grasping, GraspSplats performs object-level query, conditional part-level query, and grasp sampling. Unlike NeRF-based approaches [2], which requires costly rendering to extract language-aligned features and geometry from implicit MLPs, GraspSplats operates directly on Gaussian primitives for efficient localization and grasping queries.

**Open-vocabulary Object Querying.** We first perform object-level open-vocabulary query (*e.g., mug*), where we take language queries to select objects for grasping, with optional negative queries to filter out other objects. We do so by directly identifying 3D Gaussians whose isotropic CLIP features more closely align with positive queries over negative queries. The feature-text comparison process follows standard CLIP practices [4, 1] and is detailed in Sec. E.

**Open-vocabulary Conditional Part-level Querying.** As discussed by Rashid et al. [2], CLIP exhibits bag-of-words-like behavior (*e.g., the activation of 'mug handle'* tends to contain both mugs and handles). Thus, it is necessary to perform *conditional queries*. While LERF-TOGO [2] requires a two-step (render-voxelization) process; GraspSplats *natively supports* CLIP queries conditioned on Gaussian primitives. In particular, given an object segmented from the previous operation, we simply repeat the procedure with the new part-level query and limit the set of Gaussians to the segmented object. A qualitative example of this part-level conditioning is given in Fig. 3.

**Grasp Sampling using Gaussian Primitives.** We perform grasp sampling directly on the Gaussian primitives for streamlined grasping. To do so, we combine GraspSplats with GPG, a sampling-based grasp proposer [16, 17]. We first define a workspace $\mathcal{R}_{obj}$ as the 3D space expanded from the segmented object part. The expansion radius is the sum of both the longest axes of the scales of the Gaussian primitives and the gripper's collision radius. Then, we sample $N$ points from $\mathcal{R}_{obj}$. Within the neighborhood $R_p$ of these sampled points where $R_p$ refers to the area within a specified distance from the selected point, we aggregate Gaussian primitives with rendered normals and compute the reference coordinate system for grasp sampling with the average normal direction

$$M(p) = \sum_{g \in R_p} \hat{n}(g)\hat{n}(g)^T \tag{2}$$

| Method | Latency↓ | | Grasping Success↑ | |
|---|---|---|---|---|
| | Training | Grasping | Static | Dynamic |
| Tracking Anything [47] | — | 3.1s | 41.9% | 45% |
| ConceptGraphs* [24] | ∼30s | 0.7s | 51.1% | —† |
| LERF-TOGO [2] | ∼10min | 9.9s | 65.1% | —† |
| F3RM* [1] | ∼3min | 1.6s | 72.1% | —† |
| GraspSplats (Ours) | **60s** | 1.3s | **81.4%** | **74.2%** |

Table 1: **Comparison to NeRF and 2D-based methods on latency, static/dynamic successful rate** averaged across variations in motions and objects/parts. Latencies are reported for (1) the time for (re)building the representation; and (2) grasp latency given task texts. *: reproduced variants that use GraspNet [32] on objects segmented. †: methods require offline batch processing that does not cope with dynamic scenes. We report detailed object-/part-level successes in appendix Sec. C.

where $\hat{n}(g)$ denotes the unit surface normal of the gaussian primitive $g$. In the reference frame of each sampled point $p$, we perform a local grid search to find candidate grasps, where the finger of the gripper at terminal candidate grasps contacts the geometry of the segmented part. The details are given in Sec. G.

### 3.3 Dynamic Scene: Real-time Tracking and Optimization

Using representations optimized for semantics and geometry, it is natural to extend GraspSplats to track object displacements and edit the Gaussian primitives in real time. It is worth noting that such operation is challenging for existing NeRF-based methods [2, 1].

**Multi-view Object Tracking with keypoints.** Assume one or more calibrated cameras without egocentric motions. Given an object language query, we segment its 3D Gaussian primitives and rasterize a 2D mask to the camera. We then discretize the rendered mask into a set of points as input to a point tracker [18], which continuously tracks the 2D coordinates of given points. We translate these 2D correspondences into 3D using depth. To filter out noisy correspondences, we use a simple DBSCAN [45] clustering algorithm to filter out 3D outliers. Finally, for the remaining correspondence points, we use the Kabsch [46] algorithm to solve for the $\mathbf{SE}(3)$ transformation, which we apply to the segmented 3D Gaussians primitives. For multiple cameras, we append the estimated 3D correspondences from all cameras to the system of equations for the Kabsch algorithm. Note that the displacement can be exerted either by the arm or other external forces.

**Partial Fine-Tuning.** Edited scenes may have undesirable artifacts for regions unobserved during the initial reconstruction (*e.g.,* surface underneath displaced objects). Optionally, GraspSplats supports partial scene re-training using object masks rendered before and after the displacement, which is much more efficient than a complete reconstruction.

## 4 Experiments

In this section, we conduct experiments to validate the efficacy of GraspSplats. Specifically, our experiments aim to address the following research questions:

- Main Results — Why is GraspSplats preferable than existing NeRF- and point-based methods?

- Ablation Study — What were the design choices?

**Experiment Protocol.** We evaluate GraspSplats on several static and dynamic task settings. The full implementation details are provided in Appendix Sec. B.

- **Static zero-shot part-level manipulation.** We experiment with 8 different re-arrangements of objects (4 are cluttered, shown in Fig. 1) with 24 objects and 43 trials. The common objects with part queries including kitchenware, tableware, toys, and tools.

| Method | Segment | Grasp Sampling |
|---|---|---|
| Tracking Anything* | $2.5 \pm 0.1$s | $0.6 \pm 0.05$s |
| ConceptGraph* | **0.1** $\pm 0.05$s | $0.6 \pm 0.05$s |
| LERF-TOGO | $5.1 \pm 0.3$s | $4.8 \pm 0.7$s |
| F3RM | $1.0 \pm 0.1$s | $6.9 \pm 0.45$s |
| GraspSplats | $0.8 \pm 0.1$s | **0.5** $\pm 0.06$s |

Table 2: **Grasping latency** breakdown. standard deviations are reported over 10 runs. *: our own reproductions that use GPG [16] for grasping.

| Method | Query Time↓ | Succ.↑ |
|---|---|---|
| GraspNet-100 [2] | 10.3s | 76.7% |
| GraspNet-1 [32] | 0.6s | 65.1% |
| F3RM [1] | 6.9s | — |
| **GraspSplats** | **0.5s** | **81.4%** |

Table 3: **Query time and success rate** of different grasp sampling methods measured in the static scene. LERF-TOGO [2] uses multi-view inferences that require 100 GraspNet passes.

- **Dynamic zero-shot part-level manipulation.** Objects in this setting are similar to static grasping (40 trials on 24 objects). After the initial scan, a human operator rearranges the scene. To specifically evaluate tracking performance, we experiment with three types of dynamic motion:

  - **Easy.** Objects are translated without rotation.
  - **Medium.** Objects are rotated $180°$, with some occlusions from hand during rotation.
  - **Hard.** Objects are translated and rotated simultaneously, with certain occlusions.

## 4.1 Main Results

We demonstrate the efficacy of GraspSplats over NeRF-based methods [2, 1], 2D+Depth point-cloud based methods [47, 25] methods, and scene-graph based method [24]. We compare with two recent NeRF-based methods, LERF-TOGO [2] and F3RM [1]. Since F3RM requires human demonstrations, we implement a zero-shot variant, F3RM*, which uses GraspNet [32] to generate grasps based on rendered depth and features. For the projection-based baselines, we adapt TrackAnything [47], which combines SAM [25] and GroundingDINO [26] for segmentation. Per-frame segmented depth is used to build point clouds for input to GraspNet [32] to generate grasps. For the scene-graph-based method, we use ConceptGraphs [24] for segmentation and GraspNet for grasp generation. The results for latency, and success rate for static and dynamic scenes are shown in Table. 1. We claim the following advantages for GraspSplats:

**GraspSplats builds part-level representations more efficiently than NeRFs.** As shown in Table. 1, GraspSplats is more efficient in both representation building and grasp sampling than LERF-TOGO [2] and F3RM [1]. Besides up to 10x training speed, GraspSplats achieves better grasping success rate with 16.3 points better than LERF-TOGO, and 9.3 points better than F3RM. In addition, both baselines fall short under cluttered scenes with object overlapping without regularizing the object boundary in the reference feature.

**GraspSplats outperforms 2D foundation models**. To demonstrate how optimized representations improve grasping success, we compare GraspSplats against baselines [47, 24] that use off-the-shelf 2D vision foundation models (SAM [25] and GroundingDINO [26]). Due to existing methods' reliance on GroundingDINO [26] that is not trained on part-level data, they yield reasonable performance on object-level grasping but fail almost every part-level query.

**Our method has supreme capabilities for accurate dynamic scene modeling.** GraspSplats exploits the benefit of explicit representations, which allows editing reconstructed representations without compromising the rich geometric and semantic scene features. We demonstrate that our real-time tracking algorithm allows displacing reconstructed objects effectively, which works in dynamic scenes with a high success rate. This would otherwise be a challenging scenario for implicit methods. Additionally, our approach reconstructs scenes with greater detail than 2D-based methods, resulting in a higher success rate in dynamic scenarios.

## 4.2 Ablation Study

**Reference Feature.** We show advantages of our hierarchical features in Tab. 4. Specifically, since there are no existing scene-scale part-level segmentation datasets, we manually annotate object-

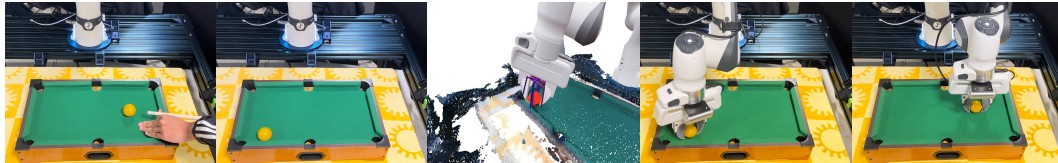

Track *yellow ball* and reset it to the serve point.

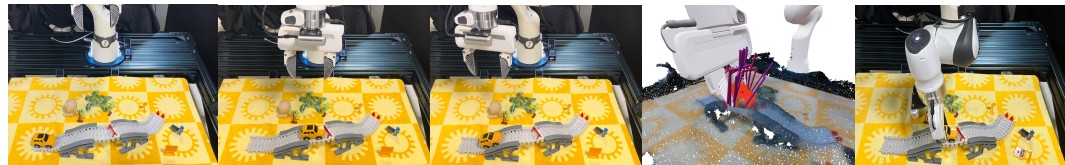

Reset *yellow truck* if it stops.

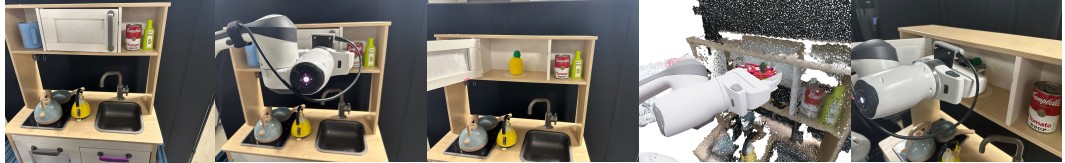

Open the *cabinet door* by the *gray handle*, and grasp the *pineapple*.

Figure 4: Qualtative examples of GraspSplats performing zero-shot task execution in real-world environments. Given object-part text queries (*italicized* in descriptions), GraspSplats executes grasping followed by heuristic trajectories. From left to right: illustration of scene change; grasp poses sampled by GraspSplats; execution of grasping. Videos can be found on the website.

level and part-level masks for 4 scenes for evaluation. GraspSplats outperforms LERF [6] on openvocabulary object-/part-level segmentation. Qualitative results are provided in Fig. 8.

**Grasp Sampling.** We ablate different grasping methods in Table. 3. GraspNet-n represents the grasps accumulated after running Grasp-Net [32] $n$ times from different viewpoints, a test-time augmentation technique used in [2]. GraspSplats demonstrates superior stability and speed in all cases. In other words, when the semantic affordance is satis-

| Method | IoU↑ |
|---|---|
| LERF [6] | 39.0 |
| **GraspSplats** | **50.7** |

Table 4: object/part IoU

factorily provided, then geometrically informed sampling based on explicit Gaussian representation yields good results. We provide further breakdown of sampling efficiency in Table. 2 and qualitative illustrations in Fig. 9.

**Qualitative Results.** Fig. 4 shows GraspSplats executing various zero-shot tasks. The heuristic policies can be easily composed to involve various object-part text queries and trajectories.

More ablation studies, such as initialization from depth, can be found in appendix Sec. C. We also present failure analysis and qualitative examples of 2D rendered heatmaps in the appendix.

## 5   Conclusion and Limitations

In this work, we present GraspSplats, a novel representation for zero-shot task-oriented manipulation. GraspSplats is efficient in building feature-enhanced 3DGS and grasp sampling. We further enhance GraspSplats with point trackers to directly edit optimized representation to capture the dynamics of objects, which would be challenging for implicit NeRF-based methods.

**Limitations.** We highlight the most pronounced limitations to facilitate future research and provide detailed failure analysis in the appendix Sec. H. The current object corresponding algorithm based on the Kabsch [46] algorithm assumes that objects undergo *rigid* transformation. While the Gaussian representation is conceptually applicable to more general deformable objects (*e.g.,* dough and clay), this is not investigated in the current work. In addition, the tracking is sensitive to fast rotation and the resulting visual occlusions and motion blurs, which could be potentially addressed as a rendering-based optimization problem.

# 6 Acknowledgement

This work was supported, in part, by NSF CAREER Award IIS-2240014.

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

| Method | Process Time↓ | Train Iteration ↓ |
|---|---|---|
| Colmap-S [12] | 11.6 | 10,000 |
| Colmap-D [48] | 623.0 | 3,000 |
| **GraspSplats** | **0.7** | **3,000** |

Table 5: Comparison of different initialization schemes. Processing time is averaged across 4 scenes. Train iteration reports the updates needed for convergence in increments of 1,000.

| Method | Object-level | Part-level |
|---|---|---|
| LERF-TOGO [2] | 81.5% | 63.0% |
| F3RM* [1] | 85.2% | 77.8% |
| **GraspSplats** | **96.3%** | **85.2%** |

Table 6: Analysis of object-level and part-level **grasping success rate** under static scenes. Note that this table is computed using 27 object-part pairs different from the runs presented in Table. 1.

## A Neural Rendering Background

Point and surface splatting methods represent a scene explicitly via a mixture of 2D or 3D Gaussian ellipsoid. In the case of Gaussian Splatting, the geometry is represented as a collection of 3D Gaussian, each being the tuple $\{\mathcal{X}, \mathbf{\Sigma}\}$ where $\mathcal{X} \in \mathbb{R}^3$ is the centroid of the Gaussian and $\Sigma$ is its covariance matrix in the world frame. This gives the probability density function

$$G(\mathcal{X}, \Sigma) = \exp -\frac{1}{2}\mathcal{X}^\top \Sigma^{-1} \mathcal{X}. \tag{3}$$

Gaussian splatting decomposes it into a scaling matrix $\mathbf{S}$ and a rotation matrix $\mathbf{R}$ via $\Sigma = \mathbf{R}\mathbf{S}\mathbf{S}^\top\mathbf{R}^\top$. The color information in the texture is encoded with a spherical harmonics map $\mathbf{c}_i = \mathrm{SH}_\phi(\mathbf{d}_i)$, which is conditioned on the viewing direction $\phi$.

To optimize for features, existing methods tend to append an additional vector $\mathbf{f}_i \in \mathbb{R}^d$ to each Gaussian, which is rendered in a view-independent manner because the semantics of an object shall remain the same regardless of view directions. The rasterization procedure starts with culling the mixture by removing points that lay outside the camera frustum. The remaining Gaussians are projected to the image plane according to the projection matrix $\mathbf{W}$ of the camera, which is then sorted from low to high using the distance from the virtual camera origin. This projection also induces the following transformation on the covariance matrix $\Sigma$:

$$\Sigma^{'} = \mathbf{J}\,\mathbf{W}\,\Sigma\,\mathbf{W}^\top\mathbf{J}^\top, \tag{4}$$

where $\mathbf{J}$ is the Jacobian of the projection matrix $\mathbf{W}$. We can then render both the color and the visual features with the splatting algorithm:

$$\{\hat{\mathbf{F}}, \hat{\mathbf{C}}\} = \sum_{i \in N}\{\mathbf{f}_i, \mathbf{c}_i\} \cdot \alpha_i \prod_{j=1}^{i-1}(1 - \alpha_j), \tag{5}$$

where $\alpha_i$ is the opacity of the Gaussian conditioned on $\Sigma^{'}$ and the indices $i \in N$ are in the ascending order determined by their distance to the camera origin.

Following the convention [38], GraspSplats assumes that per-gaussian feature vector $\mathbf{f}_i$ is isotropic. The rendered depth, images, and features are then supervised using L2 loss. Note that all recent works [38, 37, 39] follow a similar paradigm as Eq. 1.

## B Implementation Details

We calibrate cameras using Colmap [49] with Aruco visual markers. For all settings, the in-wrist camera obtains an initial scan of the scene by going in a tabletop Bezier curve. After which object-part text queries are provided. Though GraspSplats conceptually supports multi-camera tracking, we only use a single RGB-D camera in experiments. Manipulation success is defined as lifting the object (optionally, by the parts specified) for at least 3 seconds without re-attempting.

## C More Ablation Studies

**Initialization from Depth.** In Table. 5, we ablate the effect of geometric regularization by comparing several variants to initialize geometry for Gaussian. In particular, while the original Gaussian

Splatting [12] paper adopts sparse Colmap [49] initialization from RGB images (Colmap-S); some recent works have found that dense Colmap reconstruction (Colmap-D) serves as a better initialization that results in faster convergence [48]. GraspSplats adopt geometric regularization and directly initializes centers of Gaussians using depth inputs, which leads to much more efficient training.

**Success Rate Breakdown.** As shown in Table. 6, to more clearly demonstrate the comparison of part-level segmentation capabilities, we tested the success rates of 27 object-part pairs in unclustered scenes, which differs from Table. 1. It clearly shows that our approach has nearly a 100% success rate on object-level grasping while maintaining a very high success rate on part-level grasping. This is a benefit from both the feature-enhanced 3D Gaussians and part-level supervision from SAM.

## D  Part-level Features Determination

Given object-level bounding boxes, we crop and wrap images to square patches to obtain CLIP features. After per-patch features are computed, we create a part-level feature map by aggregating per-patch features using the inverse function of the cropping and wrapping process. For pixels that are assigned with more than two feature vectors, the features are averaged. Pixels with no assigned features are ignored during rendering-based feature distillation.

## E  Object-level CLIP Queries

Specifically, GraspSplats follows standard CLIP querying practices [1, 6, 2, 38] and takes a positive vocabulary with negative vocabularies. By default, the negative vocabularies include canonical words (*i.e., 'objects' and 'things'*), but can be optionally extended with user queries. To be specific, given a language set $L = \{L_0^-, L_1^-, \ldots, L_n^-, L^+\}$ containing $n + 1$ words, where $L^+$ is the positive query and the remaining $L^-$ are negative queries. The CLIP model is used to encode each word $L_i$ into a 768-dimensional feature vector $\mathbf{F}_{\text{text},i} \in \mathbb{R}^{768}$:

$$\mathbf{F}_{\text{text},i} = \text{CLIP\_encode}(L_i), \quad i = 0, 1, \ldots, n$$

For each Gaussian primitive $j$, there is a 16-dimensional view-independent feature vector $\mathbf{F}_{\text{latent},j} \in \mathbb{R}^{16}$ This is decoded into a 768-dimensional CLIP feature representation $\mathbf{F}_{\text{CLIP},j} \in \mathbb{R}^{768}$:

$$\mathbf{F}_{\text{CLIP},j} = \text{Decoder}(\mathbf{F}_{\text{latent},j})$$

Then, we calculate the cosine similarity between its CLIP feature representation $\mathbf{F}_{\text{CLIP},j}$ and each query $L_i$ in the set $L$ :

$$\text{sim}(\mathbf{F}_{\text{CLIP},j}, \mathbf{F}_{\text{text},i}) = \frac{\mathbf{F}_{\text{CLIP},j} \cdot \mathbf{F}_{\text{text},i}}{\|\mathbf{F}_{\text{CLIP},j}\| \|\mathbf{F}_{\text{text},i}\|}$$

Then, we apply a softmax function to the similarities between it and all queries $L_i$ to enhance the similarity for the positive query:

$$\mathbf{S}_j = \text{softmax}(\{\text{sim}(\mathbf{F}_{\text{CLIP},j}, \mathbf{F}_{\text{text},i})\}_{i=0}^n)$$

Here, $\mathbf{S}_j$ is the similarity vector for Gaussian primitive $j$. After applying a temperature softmax, the similarity for the positive query $L^+$ is selected:

$$\text{sim}_{\text{positive},j} = \mathbf{S}_j[n]$$

The selecting Gaussians whose similarity to $L^+$ passes a certain threshold $\tau = 0.6$ will be regarded as the grasping object. After Gaussians are selected, we apply the DBSCAN [45] clustering algorithm to filter out outliers.

## F  Reference Feature Computation

Following [14], $\mathbf{D}_{obj}$ achieves high recall by intentionally including more false positives, which ensures recall of objects. MobileSAMV2 [14] then uses $\mathbf{D}_{obj}$ as priors to generate object-level segmentation masks $\{\mathbf{M}\}$.

To generate object-level features, For a given object mask $\mathbf{M}$, we use Masked Average Pooling (MAP) to aggregate an object-level feature vector

$$w_i = \text{MAP}(\mathbf{M}, \mathbf{F}_C) = \frac{\sum_{i \in \mathbf{F}_C} \mathbf{M}(i) \cdot \frac{\mathbf{F}_C(i)}{||\mathbf{F}_C(i)||}}{\sum_{i \in \mathbf{F}_C} \mathbf{M}(i)} , \tag{6}$$

where $i$ is a pixel coordinate. We then construct $\mathbf{F}_{obj}$ by assigning $w$.

To generate part-level features, for a given bounding box, we crop and wrap the patches to (224, 224). The image patch is then processed by MaskCLIP to generate feature map of shape (28, 28, 768). We then interpolate the generated feature map to match the size of the original bounding box, and paste it onto the part-level feature map. If a pixel is assigned more than one feature, we average all assigned features.

## G  Grasp Sampling Algorithm

We define $F(p) = [v_3(p)v_2(p)v_1(p)]$ as the orthogonal reference frame at point $p$ where $v_1(p), v_2(p), v_3(p)$, correspond to the normal direction, the secondary direction, and the minimum direction of $M(p)$. We search a 2D grid $G = Y \times \Phi$. For each $(y, \phi) \in G$, we apply translations and rotations relative to $F(p)$, then push the gripper along the negative x-axis until a finger or the base of the gripper contacts the point cloud. Let $T_{x,y,\phi}$ be the homogeneous transformation describing translations in the $x, y$ plane and rotation about the z-axis. The gripper $h_{y,\phi}$ at grid cell $y, \phi$ contacts the point cloud at: $F(h_{y,\phi}) = F(p)T_{x^*,y,\phi}$, where $x^*$ is the minimum distance along the negative x-axis at which the gripper contacts the point cloud. If the number of segmented objects $N_{obj}$ within the gripper's closed region exceeds a set threshold $N_{th}$, i.e., $N_{obj} > N_{th}$, the gripper $h_{y,\phi}$ is added to the candidate grasp set $H$. Finally, we use a geometry-aware scoring model [17, 50] to rank the grasps and select the grasp pose with the highest score.

## H  Limitations and Failure Cases

We present a comprehensive analysis of limitation, or failure modes, of the current implementation of GraspSplats. We hope they will help inspire future research.

- **General Deformation.** The current scene deformation algorithm based on the Kabsch algorithm assumes that objects undergo *rigid* transformation. While the Gaussian representation is conceptually applicable to more general deformable objects (*e.g.,* dough and clay), this is not investigated in the current work.

- **Tracking Failure.** In addition, the tracking is sensitive to fast rotation and the resulting visual occlusions and motion blurs, which could be potentially addressed as an optimization problem using semantic and geometric priors fused in GraspSplats without assuming consistent object views. We illustrate a failure example in Fig. 6, where GraspSplats is tasked to track a yellow toy duck with texture-less body.

- **Manipulation Policy.** GraspSplats focuses on the grasping of the object parts via language guidance. However, in reality, it can be interesting to incorporate a more complex manipulation policy. For example, though GraspSplats supports the manipulation of articulated object, such as cabinets, by heuristic policy to pull with respect to the normal of the surface, it currently has the quasi-static assumption. Thus, it does not support more complex manipulation. Future work may explore if it is possible to learn more complex manipulation policies on top of the powerful representations of GraspSplats. In addition, though the

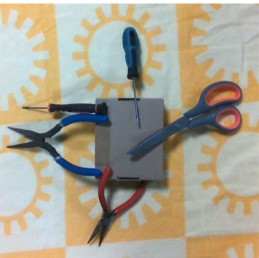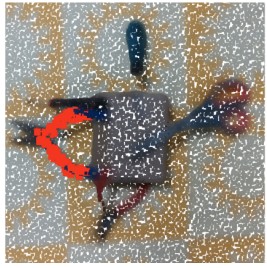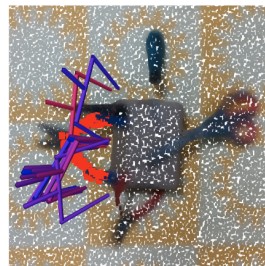

Figure 5: Grasp execution error: The leftmost image is the captured image, the middle shows the segmented grip of the pliers, and the rightmost image displays the generated grasp. The red grasp has the highest score.

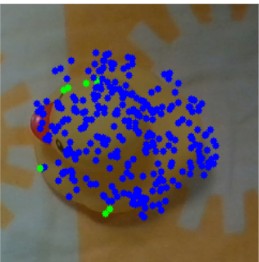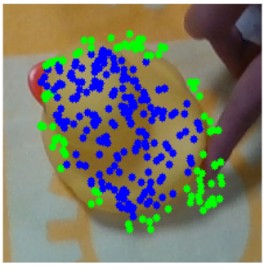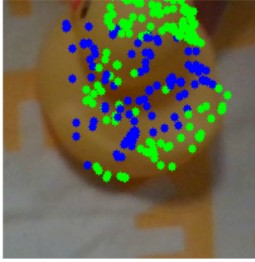

Figure 6: Tracking failure: The leftmost image shows the initially sampled feature points, with blue indicating valid feature points and green indicating lost tracking points. The two images on the right display the feature points' status as the object moves, where valid feature points are continuously lost.

       semantic design provides a good boundary between foreground and background, it does not differentiate between objects of the same types (illustrated in Fig. 5).

- **Requirement for Scene Scanning.** GraspSplats requires scanning of initial objects on the tabletop, where the scanning poses are generated from pre-programmed poses. Future work may include an active reconstruction policy to automate such a process.

## I Hardware Configuration

We use the Franka Research robot (FR3) as the main experiment platform. In addition to one Intel Realsense D435 cameras mounted on the end effector, we also use inputs from two third-person view Intel Realsense D435 cameras to facilitate scene reconstruction. We use the UMI gripper [51] as the end effector. The arm and cameras are connected to a desktop computer with a single RTX 4090 and an Intel i9-13900k CPU, on which GraspSplats is deployed.

## J Multi-state Task Policy Rollout

The examples of complex tasks are designed to illustrate potential use cases rather than showcase highly intricate operations. The tasks primarily focus on object manipulation, such as updating an object's position, grasping, resetting, and pick-and-place actions. For instance, in the toy car experiment in the supplementary video, after grasping the car, the robot resets the car by placing it back in its original position with the same orientation. During pick-and-place tasks, the placement orientation is aligned with the grasping orientation, and the placement location is determined by querying the center of another object.

These operations highlight the system's capabilities in basic object manipulation. However, challenges arise in tasks that involve precise placement, especially when the object's orientation shifts

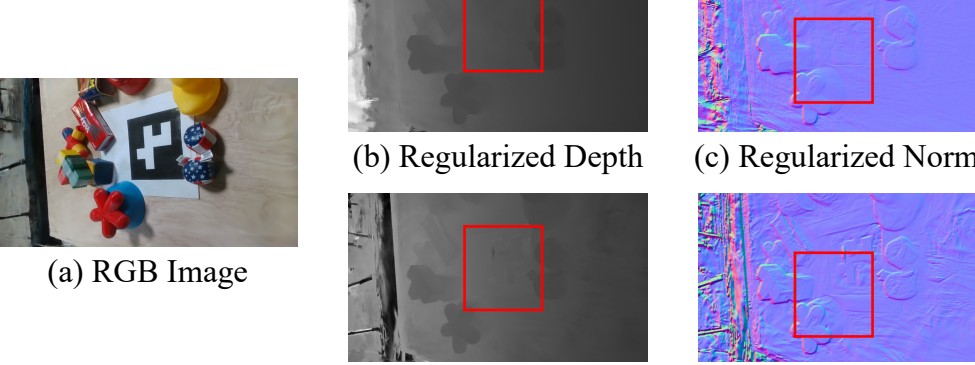

(a) RGB Image

(b) Regularized Depth

(c) Regularized Normal

(d) Unregularized Depth

(e) Unregularized Normal

Figure 7: Qualitative results of depth supervision. (a) Rendered RGB image, which is not noticeably different with or without depth supervision. (b) Rendered depth image with depth supervision. Notice the red circle surface is accurately rendered as flat. (c) Rendered normal image with depth supervision. Notice that the object edges are smoother. (d) Unregularized depth, which contains artifacts on flat patterns. (e) Unregularized normal, which is noisy near the object edge.

during grasping due to the inherent instability or rotation caused by the gripper's contact. This issue is particularly evident when the gripper obstructs the view of the object, complicating the perception of the object's orientation during placement. These challenges and potential failure cases are further discussed in the supplementary material.

## K    Effect of Depth Supervision

We have provided quantitative evidence in the paper to validate the effectiveness of the depth initialization in training. For the other aspect of geometric regularization, depth supervision during the training, we provide qualitative samples in Fig. 7.

## L    Text Query Comparison

The 2D response maps for the same text query generated by different algorithms are displayed in Fig. 8. Grounded SAM[52] in TrackAnything[47] demonstrates effective segmentation at the object level; however, it fails to distinguish parts of objects and sometimes does not respond at all. Additionally, it tends to merge multiple closely positioned objects into one. LERF[6] similarly exhibits weak responses at the part level, with very unclear segmentation between objects, making it nearly impossible to separate similar objects that are close together. F3RM[1] provides higher-quality features compared to LERF and can respond to some part-level queries, but it also struggles with accurately distinguishing between similar, closely positioned objects. In contrast, our algorithm exhibits near 2D segmentation capabilities and is highly sensitive to part-level queries, allowing for precise differentiation of different parts within distinct objects.

## M    GraspNet vs Grasp Sampling

In our comparative analysis of grasp sampling methods, we employed GraspNet[32] with collision detection as LERFTOGO[2], sampling viewpoints on a hemisphere defined by theta and phi parameters. Specifically, we tested configurations with 3x3, 5x5, and 10x10 viewpoints, effectively running GraspNet 9, 25, and 100 times respectively. The resulting grasps for these configurations are illustrated in the first three images as is shown in Fig. 9. Notably, the red circles highlight regions

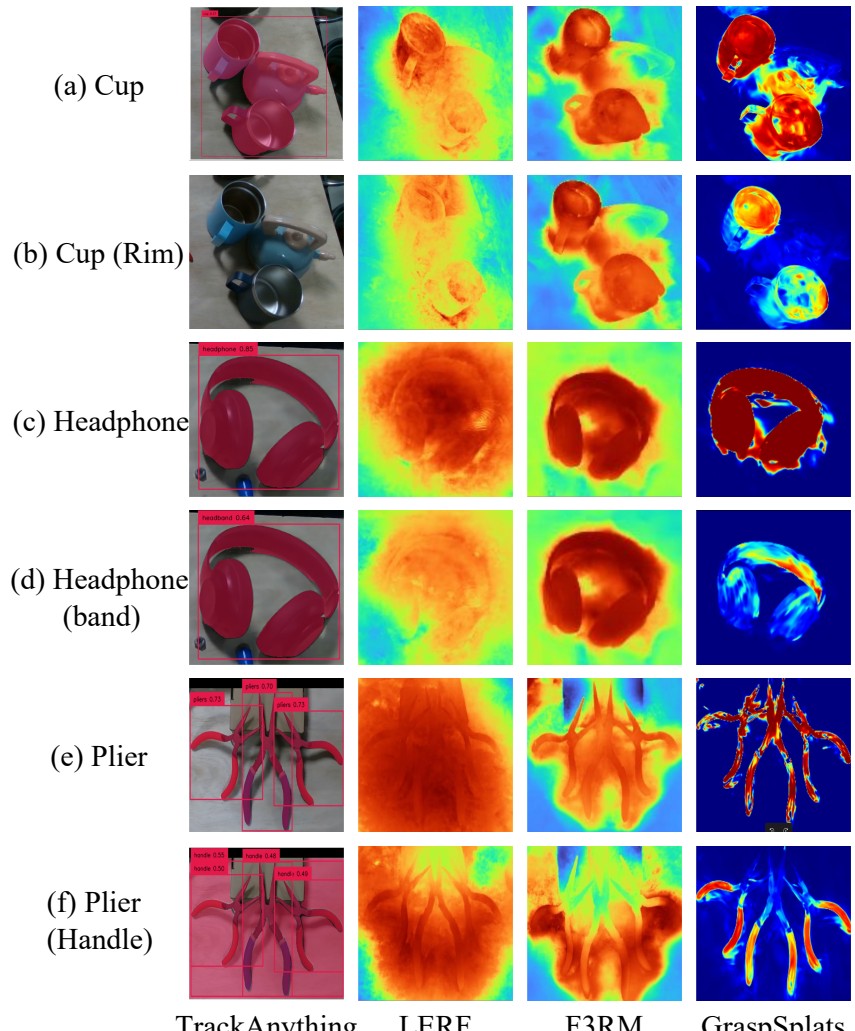

|  | | | |
|---|---|---|---|
| (a) Cup | | | |
| (b) Cup (Rim) | | | |
| (c) Headphone | | | |
| (d) Headphone (band) | | | |
| (e) Plier | | | |
| (f) Plier (Handle) | | | |
| TrackAnything | LERF | F3RM | GraspSplats |

Figure 8: From left to right are the responses of TrackAnything (Grounded SAM), LERF, F3RM, and our algorithm to the same text query.

| Method | Query Time↓ |
|---|---|
| GraspNet-100 | 10.3 |
| GraspNet-25 | 3.2 |
| GraspNet-9 | 1.2 |
| Ours | **0.5** |

Table 7: Time efficiency comparison of different grasp sampling methods including ours and GraspNet with different number of executions.

where valid grasps were not generated. In contrast, for our sampling-based method, we configured the system to sample 3000 points within the workspace, utilizing 16 threads of 12900K for parallel processing. The outcome of this approach is depicted in the last image. We also compare the time efficiency of the algorithms as is shown in Tab. 7

It is worth noting that while the GS grasps are directly generated with Gaussians, the point cloud data (pcd) is used solely for visualization purposes. This comparison underscores the efficiency and potential limitations of each method in terms of grasp generation and computational demands.

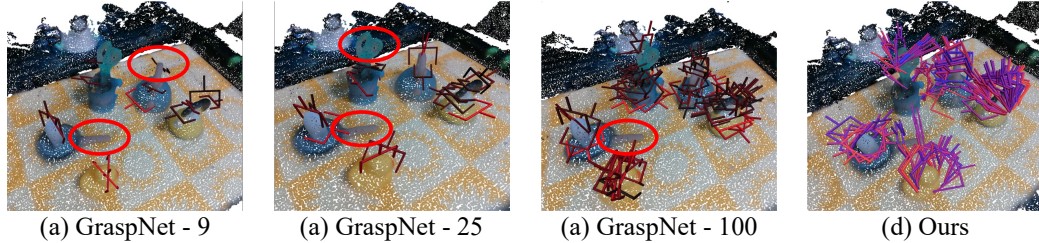

| (a) GraspNet - 9 | (a) GraspNet - 25 | (a) GraspNet - 100 | (d) Ours |

Figure 9: The first three images from left to right show the grasps obtained by running GraspNet with 9, 25, and 100 different viewpoints, respectively. The last image shows the results obtained by our method, which sampled 3000 points in the workspace.

