# OpenReview forum: "GraspSplats: Efficient Manipulation with 3D Feature Splatting"
_robot-learning.org/CoRL/2024/Conference — CoRL 2024_

### Official Review · Reviewer_WC1J · 2024-07-19
**Sound main idea, but needs some improvements.**

**Originality:** 3
**Technical Quality:** 4
**Clarity Of Presentation:** 4
**Potential Impact:** 2
**Recommendation:** 3
**Confidence:** 3

**Review:**

Overall, the idea looks sound and straightforward. The main novelty is using Gaussian splats for modeling language-embedded feature fields instead of neural radiance fields. The paper's main claim is that Gaussian splats are superior for distilling feature fields, and the experiments demonstrate this claim both in speed and performance. However, I have the following questions and concerns:

* I understand that Gaussian splats are faster than Nerfs, but I don’t understand why they perform better regarding learned features, as shown in Table 2. This table shows that somehow, gaussian splats learn better dense visual features than Nerf (As I assume, prediction of F_part and F_obj should be similar). Moreover, in lines 241-242, the paper mentions that Nerf-based methods perform better on static scenes.
* Grasp success is only estimated using whether the robot can lift the object for 3 seconds. This means the grasp success results do not consider the grasped part. However, one of the main contributions of the work was task-oriented grasping.
* In the supplementary video, complex language queries correspond to multistep plans. How did you process language queries such that these plans are generated? For example, in one of the tasks (toy car experiment in 1.58), the robot picks up the car when it stops and puts it in its initial position. However, I expect this method only to handle the car grasping part. How is the rest accomplished? The paper does not discuss these.
* The results in Table 2 are only from 4 scenes, which is relatively low.
* The experiment sections contain a very minimal qualitative analysis of the method. I noticed that most of these analyses (also limitations) are included in the supplementary video and material. Some of these should be moved to the main paper.

Clarity:

There are some clarity issues on paper:
* In equation 1, N, as I understand it, is the number of Gaussians in the Gaussian Splat, but this is not discussed anywhere.
Lines 148 to 152 explain how F_part is estimated, but it is quite unclear. what does uniform query mean? Why is N significantly smaller than the number of patches needed from uniform queries? In line 151, how F_part is estimated is very ambiguous (which features are you talking about in this line?). It would be better to write this part more mathematically for easy understanding.
* Why do you use cosine loss to train F_part and F_object MLPs?
* Between lines 169-170, how queries are selected is unclear. Maybe you can write this part more mathematically as well.
* The Open-vocabulary Conditional Part-level Querying section (Lines 172-178) discusses how part queries lead to activation of both the object and the part (e.g. object and part). To prevent this, you segment the object; however, why does this prevent activation of the object as it still appears in the image?

Other Comments:
* The paper discusses that it is not possible to model dynamic scenes with implicit representations. While I agree it is better to model dynamic scenes with explicit methods, this is not exactly true. For example, it would be quite easy to apply rigid transformations to the SDF-based methods. Furthermore, many works model dynamic scenes with Nerfs using deformation fields (Some examples are [1], [2]).
* There is a line of work in robotics that predicts grasp poses based on descriptors or dense correspondences [3-7]. This line of work is very relevant to this paper, as they also use such representations for grasp prediction.

References:

[1] Park, Keunhong, et al. "Nerfies: Deformable neural radiance fields." Proceedings of the IEEE/CVF International Conference on Computer Vision. 2021.

[2] Lin, Chen-Hsuan, et al. "Barf: Bundle-adjusting neural radiance fields." Proceedings of the IEEE/CVF international conference on computer vision. 2021.

[3] Florence, Peter R., Lucas Manuelli, and Russ Tedrake. "Dense object nets: Learning dense visual object descriptors by and for robotic manipulation." arXiv preprint arXiv:1806.08756 (2018).

[4] Manuelli, Lucas, et al. "kpam: Keypoint affordances for category-level robotic manipulation." The International Symposium of Robotics Research. Cham: Springer International Publishing, 2019.

[5] Simeonov, Anthony, et al. "Neural descriptor fields: Se (3)-equivariant object representations for manipulation." 2022 International Conference on Robotics and Automation (ICRA). IEEE, 2022.

[6] Yen-Chen, Lin, et al. "Nerf-supervision: Learning dense object descriptors from neural radiance fields." 2022 international conference on robotics and automation (ICRA). IEEE, 2022.

[7] Sucar, Edgar, Kentaro Wada, and Andrew Davison. "NodeSLAM: Neural object descriptors for multi-view shape reconstruction." 2020 International Conference on 3D Vision (3DV). IEEE, 2020.

**Quality Of The Limitations Section:**

3

**Questions For Rebuttal:**

My main concerns are:

* As I see there is no quantitative analysis part-level grasping.
* The paper requires some clarity improvements.

Other than that, the authors can also address other comments on my review.

Update: My concerns were addressed.

**Robotics Focus:**

4

**Summary Of Paper:**

This method proposes Gaussian splatting-based language-embedded feature fields for task-oriented grasping in dynamic scenes. Given the scene, the method first constructs a GaussianSplat that models the scene in terms of color, depth, and dense visual features. The features are then passed to shallow MLPs to predict object and part features. These features are used together with language-based queries to find the corresponding grasp region on the scene. From this region, grasp samples are generated through an antipodal grasp sampler (GPG). These grasp samples are ranked using geometry-aware scoring methods, and the best grasp is selected. Additionally, the paper uses a tracker to update the underlying feature field by rigidly transforming the Gaussians of the GaussianSplat. This makes the model robust to the changes in the scene.

**Summary Of Recommendation:**

I think paper is well written and demonstrated. However, core idea is using gaussian splatting instead of Nerfs for learning feature fields. It is not a significant novelty.

---

### Official Review · Reviewer_vXka · 2024-07-21
**Review for GraspSplats: Efficient Manipulation with 3D Feature Splatting**

**Originality:** 2
**Technical Quality:** 4
**Clarity Of Presentation:** 3
**Potential Impact:** 3
**Recommendation:** 3
**Confidence:** 5

**Review:**

**Strengths**
- The paper is well-written and easy to follow. The related work is comprehensive and the problem formulation is clear.
- The proposed hierarchical feature extraction leverages object masks to get part-level features. This is an improvement over hierarchies of multi-scale crops which is both expensive to extract and distill. Both object and part-level features are distilled into the Gaussian primitives.
- Extensive real-world experiments and ablation studies to demonstrate the robustness of the proposed approach.
- The appendix and video is useful for showcasing the results. The demos with the object being moved around are impressive.

**Weaknesses**
- The explanation for the grasp sampling in Section 3.2 was not clear to me. How do you get $R_p$ from the sampled points and what is $M$? It seems like this formulation is based on GPG but it would be good to keep this self-contained within this paper or explain it through a figure.
- Limited information on the Partial Fine-Tuning in Section 3.3 and how it works. Are additional Gaussians potentially added here?
- The object tracking is inherently based on the results of a 2D point tracker on a single view. This means that tracking will fail if objects are rotated in such a way that the points become out of view or the object becomes occluded.

**Quality Of The Limitations Section:**

3

**Questions For Rebuttal:**

See issues raised in the weaknesses above.

**Other questions**
1. One advantage of neural fields is being able to use RGB images and reconstruct transparent and specular objects. Did you experiment with such objects? Does using RGB-D and depth supervision degrade these properties?
2. For moving the Gaussians as objects move dynamically, could you further optimize them based on the incoming image observations? How do you think that would compare to using the 2D tracker?

**Robotics Focus:**

4

**Summary Of Paper:**

The paper introduces GraspSplats, an approach for object and part-level language-guided grasping that leverages 3D Gaussian Feature Splatting as the underlying scene representation. By using 2D point trackers, the Gaussians can be dynamically updated as objects are moved around in the scene. This is a significant advantage compared to implicit representations including NeRFs. The authors additionally show that the normals of the Gaussians can be used for real-time antipodal grasping. Extensive results in the real world demonstrate the improved speed of GraspSplats and its ability to handle dynamic scenes of varying levels of difficulty.

**Summary Of Recommendation:**

While the paper presents several incremental improvements to Feature 3DGS, these improvements are critical for moving towards real-world deployability. The ability to enable dynamic scene updates and real-time antipodal grasping is a significant improvement over NeRF-based representations

---

### Official Review · Reviewer_jdjs · 2024-07-25
**GraspSplats Paper - Official Review**

**Originality:** 3
**Technical Quality:** 3
**Clarity Of Presentation:** 3
**Potential Impact:** 3
**Recommendation:** 3
**Confidence:** 4

**Review:**

### Strengths

- Using 3D Gaussian splatting for dynamic scene representation is not completely novel, see [1]. The way this work approaches it, the fact that they handle object parts, and the nice experimental results make this work novel enough and significant for the community.
- The proposed algorithm seems to work well in dynamic environments which are typically hard and a limitation of previous approaches.
- The authors provide a qualitative video with real robot demonstrations for an extensive and varied set of tasks that seem complex enough.
- The authors provide further qualitative study in the supplementary material. Including depth supervision effects, qualitative comparison of text queries for different approaches, GraspNet vs Grasp Sampling, and an extended study of the real robot failure cases.

### Weaknesses

- Figure 2 and Figure 4 are not very clear and hard to understand. I believe that these figures might be more useful and help to understand the method better if they would show where/how MobileSAMV2 and MaskCLIP come into play, and details are missing.
- The paper claims that the proposed representation should be better than explicit (voxel or point) representation but they haven’t provided any comparison to any baselines using point clouds or voxel. I believe that this comparison is needed to motivate the use of GraspSplats.
- A comparison to [1] or at least a citation to this work is missing.
- In Section 4.2, the author presents a comparison of part-level segmentation against LERF. However, I don’t think this comparison is very strong since they evaluate only 4 scenes. A comparison on another dataset (e.g., [1]) or a bigger number of annotated scenes is necessary.
- Details for the experimental setup are missing: How many trails do the authors carry out per type of dynamic motion (easy, medium, hard)?
- Real robot video comparisons of the other baselines are missing (Tracking Anything, LERF).

[1] Object-Aware Gaussian Splatting for Robotic Manipulation, ICRA24 Workshop 3D Manipulation, Li et al.
[2] PartNet: A Large-Scale Benchmark for Fine-Grained and Hierarchical Part-Level 3D Object Understanding, CVPR19. Mo, et al.

**Quality Of The Limitations Section:**

3

**Questions For Rebuttal:**

- Improve the clarity of Figures 2 and 4.
- Improve the experimental part including explicit (voxel and/or point-cloud) representations.
- Add the missing information in the experimental section and improve the Table 2 results by adding a more significant experiment/dataset.
- Add grasping qualitative comparison to baselines.

**Robotics Focus:**

4

**Summary Of Paper:**

The authors propose GraspSplats, a zero-shot object part manipulation algorithm based on 3D Gaussian Splatting. The proposed algorithm takes RGB-D frames from a calibrated camera and reconstructs the scene using 3D Gaussian ellipsoids. The 3D Gaussians are initialized from the depth frames, and GraspSplats uses MobileSAMv2 and MaskCLIP to compute object and part-level features. To handle changes in the scenes, GraspSplat uses multi-view object tracking. Then, zero-shot language-guided manipulation is performed by supplying the object and part name that needs to be manipulated to generate CLIP queries to identify the 3D Gaussians that better aligns. Then, a sampling-based grasp proposer is leveraged to compute candidate grasps around the . GraspSplats demonstrated to be successful in real robot experiments, and the paper presents a comparison of its performance against Tracking Anything, LERF-TOGO, and a zero-shot variant of F3RM in both static and dynamic scenes.

**Summary Of Recommendation:**

I am in between weak accept and weak reject, but I am currently leaning towards weak reject. I believe that the robotics community could benefit from this paper's findings, especially for object-part manipulation. However, I am concerned that the experimental part is not strong enough as detailed in my review. |||| Update after rebuttal: After reading other reviewers' comments and the rebuttal, I will raise my rate to Weak Accept. Authors have clarify and addressed most of my concerns and I am convinced by the answer given to the other reviewers.

---

### Official Review · Reviewer_s84c · 2024-07-31
**GraspSplats Review**

**Originality:** 2
**Technical Quality:** 3
**Clarity Of Presentation:** 3
**Potential Impact:** 2
**Recommendation:** 2
**Confidence:** 3

**Review:**

To the reviewer's understanding, the novelty is in a) the inclusion of both object-level and part-level features, b) the use of an external tracker to enable language-guided tracking, and c) integration with a grasping proposal method. To the reviewer's understanding, the usage of depth data to seed the GS is relatively common and not a significant modification.

The evaluation should compare to the cited paper [33], as a baseline that also utilizes a) utilizes GS for grasping, and b) can incorporate changes quickly (relatively speaking). In fact, an obvious baseline of re-seeding the GS from the current representation upon a change is missing. Even if not real-time, this is an important comparison to make.

A strength of this work is their proposed separation of object-level vs. part-level reasoning. More detailed results on how this distinction results in improved part-level tracking would strengthen the paper.

An important limitation of the proposed tracking method is the need for a language query ahead of time to know which object will move in order to identify the keypoints to track. In the real world, multiple objects may be moving or the object that is moving may not be known a priori. Perhaps a method to automatically identify and track motion would help avoid this issue.

In the experiments, it would be good to separate out results in Tables 1 and 2 into object-level and part-level results. This helps identify in which scenarios the proposed adjustments really help. Additionally, since efficiency is a major selling point, the timing results throughout this paper should be much more precise - they currently read more like rough estimates.

The use of the word "deformation" here seems misplaced - this rather evokes deformable objects (i.e. non-rigid). Considering the tracking explicitly assumes rigidity, this is especially misleading.

**Quality Of The Limitations Section:**

2

**Questions For Rebuttal:**

How many grasps are attempted for the results shown in Tables 1 and 3? How many comparisons are made in Table 2? What is the breakdown of performance of each method between object-level and part-level grasping?

What are the variance in grasp timing for Table 1? The timing numbers seem like estimates, but if inference time is an important part of the value of the proposed method, there should be more clarity. What is the breakdown of timing w.r.t. subcomponents like finetuning the representation or evaluating grasp proposal?

Is the partial fine-tuning presented in Sec. 3.3 used in experiments?

The explanation of how part-level features are determined is unclear - how do you perform interpolation to go from labels at parts to labels across the scene?

**Robotics Focus:**

4

**Summary Of Paper:**

This paper proposes GraspSplats, a system for utilizing Gaussian Splatting (GS) for grasping. They modify GS for grasping, adding joint object-level/part-level reasoning, integration with an external tracker to handle object motions, and add a sampling-based grasp method using the resulting representation. They compare to existing language-guided zero-shot grasping methods on a real system and show favorable results.

**Summary Of Recommendation:**

The proposed work seems well motivated and shows strong performance, but is missing some important details in the experiments and should compare to existing grasping methods with GS. Additionally, their tracking relies on language guidance before hand, which seems to severely limit the applicability of the proposed tracking method. More baselines and clarity in experimentations and addressing this limitation would strengthen the work.

---

### Author Rebuttal · Authors · 2024-08-14

### Overview:

We have carefully addressed all reviewer comments directly in the revised manuscript, with responses updated in the review comments and relevant links included. This submission primarily consists of the revised paper, along with supplementary result images and videos that further illustrate our method's performance. We sincerely thank the reviewers for their valuable feedback.

### Contents:

1. **concept_graphs.png**:

    This file contains the qualitative analysis used for comparison with the pointcloud-based baseline algorithm in our experiments.

2. **figure2.png, figure3.png, figure4.png**:

    These are the revised figures from the manuscript, updated to more clearly present the pipeline and other key aspects of our methodology.

3. **fast_grasping.MOV**:

    A video example demonstrating the fast grasping capabilities of our method, highlighting the algorithm's efficiency.

4. **MultiTracking.mp4**:

    A video example showcasing the multi-target tracking and grasping capabilities of our method, demonstrating the algorithm's performance in handling dynamic scenes.

5. **revised_paper.pdf**:

    The updated manuscript, which includes both the main paper and the appendix. All modifications are highlighted in yellow and address the reviewers' comments comprehensively.

---

### Decision · Program_Chairs · 2024-09-04

**Decision:**

Accept

**Comment:**

The reviewers are in agreement that this paper addresses an important topic, is relevant to the conference, and provides some novelty in the form of extracting hierarchical features in Gaussian splat representations. The reviewers also generally agree in the strength of the contribution and it providing interesting results to the robot learning community.

However, reviewers also have a variety of concerns which the authors should address in the rebuttal / discussion phase.
Those that jumped out at me as most important are:
- The lack of evidence to support the claim that the approach outperforms purely point-based or voxel-baesd approaches
- The potential fragility of the tracker as an important limitation and discussion of how this could be overcome
- Issues with clarity in the presentation

Following the rebuttal I think the improved experiments to compare to a point-based representation and the additional tracking results address the majority of the reviewers concerns. While some might argue about the level of novelty of the paper, I think it provides an interesting investigation that would be of utility to the robot learning community.